# Constructing Randomly Lamellar HKUST–1@Clinoptilolite through Polyethylene Glycol—Assisted Hydrothermal Method and Coordinated Complexation for Enhanced Adsorptive Separation for CO_2_ and CH_4_

**DOI:** 10.3390/nano13121860

**Published:** 2023-06-14

**Authors:** Mingxuan Zhang, Jiawei Zhou, Chunlei Wan, Ming Liu, Xia Wu, Jihong Sun

**Affiliations:** Beijing Key Laboratory for Green Catalysis and Separation, Department of Chemical Engineering, Beijing University of Technology, Beijing 100124, China

**Keywords:** clinoptilolite, HKUST-1, composites, fractal structures, delamination, adsorptive separations

## Abstract

Clinoptilolite (CP) was successfully synthesized via a hydrothermal route in the presence of polyethylene glycol (PEG), and it was then delaminated by washing using Zn^2+^ containing acid. HKUST-1, as one kind of the Cu-based MOFs, showed a high CO_2_ adsorption capacity owing to its large pore volume and specific surface area. In the present work, we selected one of the most efficient ways for preparing the HKUST-1@CP compounds via coordination between exchanged Cu^2+^ and ligand (trimesic acid). Their structural and textural properties were characterized by XRD, SAXS, N_2_ sorption isotherms, SEM, and TG-DSC profiles. Particularly, the effect of the additive PEG (average molecular weight of 600) on the induction (nucleation) periods and growth behaviors were detailed and investigated in the hydrothermal crystallization procedures of synthetic CPs. The corresponding activation energies of induction (*E_n_*) and growth (*E_g_*) periods during crystallization intervals were calculated. Meanwhile, the pore size of the inter-particles of HKUST-1@CP was 14.16 nm, and the BET specific area and pore volume were 55.2 m^2^/g and 0.20 cm^3^/g, respectively. Their CO_2_ and CH_4_ adsorption capacities and selectivity were preliminarily explored, showing 0.93 mmol/g for HKUST-1@CP at 298 K with the highest selective factor of 5.87 for CO_2_/CH_4_, and the dynamic separation performance was evaluated in column breakthrough experiments. These results suggested an efficient way of preparing zeolites and MOFs composites that is conducive to being a promising adsorbent for applications in gas separation.

## 1. Introduction

Clinoptilolite (CP), as one of the heulandite (HEU) zeolites [1], is widely used for gas adsorption, industrial catalysis, and environmental treatment [2,3,4,5,6] due to its adjustable acidity, thermal stability, and unique micropore networks. It consists of channel A (0.44 × 0.72 nm) with 10-member rings, and channel B (0.40 × 0.55 nm) with 8-member rings that is parallel to channel A, as well as C (0.41 × 0.47 nm) with 10–member rings that intersect with both channel A and B. CP is widely applied in CO_2_ and CH_4_ separation because of its low cost, high productivity, large adsorption, and easy purification [7].

The synthesized CPs with the HEU structures presented the layered accumulations containing the ordered and expandable lamellar structures [8], which are conducive to not only improving the mass transport efficiency for the guest molecules but also for providing the more accessible surfaces adsorption sites for functional assembly. However, the steric hindrances of their HEU structures, caused by small interlayer spaces, strongly limit their wider applications. In this regard, we proposed a synthesis method of adding a variety of surfactants to the synthesis system for facilitating the realization of layered structure expansions and the lamellar disorder arrangements of the synthesized CPs [9,10]. As reported by Iwakai et al. [11], the silicate-1 with smaller particle size (about 40~60 nm) could be synthesized by the addition of polyoxyethylene polyether in the conventional hydrothermal system, while the particle size of conventional silicate–1 was larger than 500 nm. Silva et al. [12] provided a synthesis method of zeolite Y by adding surfactant cetyltrimethylammonium bromide in the hydrothermal system. Compared with commercialized USY zeolite, their products showed the same structure characteristics with relative lower crystallinity (about 33% of USY zeolite) but a much higher external surface area (661 cm^3^/g) than that of USY zeolite (208 cm^3^/g). Koohsaryan and Anbia proposed an efficient way of synthesizing zeolite 13X via additive polyethylene glycol (PEG), which was beneficial to synthesize the well-developed zeolite crystals that have a mesopore size (around 2–10 nm) with a shorter growth period and a relative greater crystallinity [13]. Therefore, we believe that the polymer polyols are beneficial to the layered expansion of synthetic CPs, owing to their good water solubility and greater amount of hydrogen bonds in aqueous solution.

The traditional method of zeolite delamination is usually carried out under ultrasonic conditions. Corma et al. [14] reported an exfoliated method for layered ITQ-2 by cetyltrimethylammonium bromide and tetrapropyl ammonium hydroxide under ultrasonic condition. After exfoliation, the sheets of the zeolite became a random arrangement with an external surface area of about 700 m^2^/g, almost twice as much as the raw ITQ-2. However, the high energy cost of the ultrasonic process of zeolites is one shortcoming. Gorgojo et al. [15] achieved a direct exfoliation method by cetyltrimethylammonium ion without ultrasonic treatment, and the exfoliated Nu-6 presented the larger surface areas of around 300 m^2^/g, six times that of raw Nu-6. As demonstrated by Okrut et al. [16], the layered silicate magadiite was delaminated via Zn^2+^ modification and acid treatment without ultrasound under mild conditions, resulting in that its external surface area enlarging from 10 to 25 m^2^/g.

The delamination process usually follows the synthesis process, so combining two processes may improve the delamination performance. Roth et al. [17] proposed a delamination method by calcinations based on the as-synthesized MCM-22 with additive hexamethylenediamine, showing larger *c*-axis distance from around 0.25 nm to larger than 0.50 nm. However, rare reports proposed delamination methods regarding the abovementioned CP. Hopefully, the delamination of the synthetic CPs can not only open the lamellar structures from one direction of the crystal planes but also improve the adsorption capacity and the catalytic performance of the gases.

As is well known, the metal-organic frameworks (MOFs), as kinds of multi–dimensional structure materials with high specific surface areas and abundant functional sites [18], have recently presented strong potential applications in adsorption, separation, and catalysis. HKUST-1, consisting of two channels, namely, a 0.9 nm cage and a 0.35 nm pore [19], exhibited a high CO_2_ adsorption capacity (up to 4.20 mmol/g at 27 °C and 1 bar) owing to its large pore volume (0.69 cm^3^/g) and specific surface area (1615 m^2^/g) [20], but large-scale production and industrial application are difficult to achieve due to its low productivity, poor hydrothermal stability, and higher cost. In order to avoid these disadvantages, at least to some extent, one of the most efficient methods is the preparation of the composites of zeolites and MOFs. The composites are usually prepared by such a method that the MOFs are synthesized from the precursor solution with the addition of certain amounts of the synthesized zeolites. For example, the core-shell structured Zeolite-5A@MOF-74 was obtained from the MOF-74(Ni) precursor solution with the addition of Zeolite-5A, showing higher thermal stability [21]. The CO_2_/H_2_, CO/H_2_, CH_4_/H_2_, and N_2_/H_2_ selectivity was 8659, 2375, 114, and 45, respectively, at 25 °C and 1 bar, higher than that of Zeolite–5A or MOF-74. Using a similar method, MIL-100 (Fe)@SBA-15 was successfully synthesized by Mahmoudia et al. [22], and the adsorption efficiency for methylene blue, rhodamine B, methyl orange, and alizarin yellow R dye solutions was higher than that of pure MIL-100 (Fe), due to the generations of the pore structures in the MIL-100 (Fe). As demonstrated by Tari et al. [23], the MCM-41/Cu(BDC) was prepared via the same method mentioned above, showing a higher CO_2_/CH_4_ selectivity (the separation factor of around 4.9 at 303 K and 4 bar) as compared with pure MCM-41. In addition, transition metal cations can be introduced onto CP through ion exchange, which can not only balance the anions of the aluminosilicate frameworks but also provide the central ions for preparing MOFs. However, the composites between CP and MOFs are rarely reported. More recently, we proposed a novel constructed strategy of the disorderly layered UiO-66-on-clinoptilolite heterostructures through the assistance of PEG and polyvinylpyrrolidone, and the prepared hybrid materials exhibited an excellent adsorption selective performance for CO_2_ and CH_4_ [9].

Herein, we developed a preparation method of the composites of CP and HKUST-1 via delamination, ion exchange, and coordination. On the basis of the abovementioned PEG-assisted hydrothermal method [9] and using alkali, aluminum, and silicon sources as raw materials, the synthesized CP was delaminated in the Zn-containing HNO_3_ solutions, and then Cu^2+^-exchanged after ammonization. Finally, the obtained Cu–CP was coordinated with trimesic acid ligand to form HKUST-1@CP composite without additional metal sources. Meanwhile, X-ray diffraction (XRD), Fourier transform infrared (FT-IR) spectra, thermogravimetric (TG) analysis, N_2_ sorption isotherms, and scanning electron microscopic (SEM) images were used to elucidate the structure features and physicochemical performances of the resultant HKUST-1@CP composites. Meanwhile, the application of the composites in the CO_2_/CH_4_ adsorptive separation and the column breakthrough performance were preliminarily explored. The novelty and contribution of the present work are the successful preparation of HKUST-1@CP composites by stratification, ion exchange, and coordination, especially the addition of PEG in the hydrothermal synthesis system being conducive to the layered expansion of CPs. The other contribution is that the exchanged Cu^2+^ located in the exchange position of CP can be used not only as cations to balance the negative charge of CP skeleton but, also, as central ions to complex with the ligands. As an effective adsorbent, hopefully, the CO_2_/CH_4_ adsorptive separation and the column breakthrough performance could be enhanced.

## 2. Experimental Methods

### 2.1. Materials

NaOH (99.0 wt%), KOH (85.0 wt%), Al(OH)_3_ (99.0 wt%), NH_4_Cl (99.5 wt%), CuCl_2_·2H_2_O (99.0 wt%), Zn(NO_3_)_2_·6H_2_O (99.0 wt%), HNO_3_ (68.0 wt%), and triethylamine (99.5 wt%) were obtained from Fuchen Chemical Reagent Co., Tianjin, China. Aqueous colloidal silica sol (Ludox JN–30 1.2 g/cm^3^ 30 wt% SiO_2_) was provided by Qingdao Ocean Chemical Reagent Co., Shandong, Qingdao, China. PEG (average molecular weight of 600), ethanol (99.7 wt%), and trimesic acid (98.0 wt%) were purchased from Shanghai Aladdin Biotechnology Co., Shanghai, China. All of these chemicals were analytical grade. Natural CP was sieved in 400 meshes as seed crystals. Deionized water (resistivity of 18.25 MΩ·cm, 298 K) was supplied by Zhiang–Best Water Purifier, Shanghai, China.

### 2.2. Synthesis of CP

Firstly, NaOH, KOH Al(OH)_3_ and deionized water were mixed and transferred into a Teflon-lined stainless steel autoclave, and then stirred at 150 °C for 3 h to get an alumina sol. Subsequently, silica sol, deionized water, 3 wt% of seed, and 7 wt% of PEG were mixed and added into the alumina sol. The molar ratios were as follows: 1.38 Na_2_O: 1.38 K_2_O: 11.18 SiO_2_: Al_2_O_3_: 294 H_2_O. The mixture solution was stirred at room temperature for 2 h and continually transferred into a Teflon-lined stainless steel autoclave at 150 °C for 72 h to crystallize. Afterwards, the products were washed with enormous amounts of deionized water and dried at 120 °C overnight, and named CP–X, where X is the crystallization time (h). In comparison, pure CP was synthesized without any additive PEG.

### 2.3. Delamination, Ammonization and Ion Exchange of CP-X

Firstly, 0.5 g of the synthesized CP-X was added into 30 mL of 0.5 mol/L Zn(NO_3_)_2_ solution, and after stirring for 2 h at room temperature, 15 mL of 2 mol/L HNO_3_ solution was added and then stirred for 1 h continually.

After being washed with deionized water and dried at 120 °C for 12 h, the obtained products were delaminated, and named CP-X-E. Next, the delaminated CP-X-E was added into 1 mol/L NH_4_Cl solution and stirred at 80 °C for 2 h, following: 1 g CP-X-E: 100 mL NH_4_Cl solution. After being washed with deionized water and then dried at 120 °C overnight, this procedure was repeated 10 times, and the final product was named Ammonized CP-X-E.

The ammonized CP-X-E was then added into CuCl_2_ solution with a concentration of 0.0005, 0.003, and 0.06 mol/L, and then stirred at 80 °C for 2 h, following: 1 g ammonized CP-X-E: 200 mL CuCl_2_ solution. After being washed with deionized water and dried at 120 °C overnight, this procedure was repeated 1–10 times, and the final product was named Cu-CP-X-E-Y, where Y represents the Cu content (wt%) in the samples. For example, when the X (the crystalline time) equals 72 h, the Y equals 7.3, and the corresponding sample was named Cu-CP-72-E-7.3. Various Cu-CPs and their copper contents are summarized in Appendix A of the Appendix A.

### 2.4. Synthesis of HKUST-1

HKUST-1 was synthesized on the basis of the reported procedure [24] as follows: 1.539 g CuCl_2_·2H_2_O and 1.26 g trimesic acid were dissolved into 75 mL deionized water and 75 mL ethanol for stirring at 50 °C, and, after that, 2.5 mL triethylamine was added into the above solution and stirred for 3 h continually. The powder was cooled down and washed with ethanol to remove unreacted reactant, and it then was centrifuged and dried at 120 °C overnight.

### 2.5. Synthesis of HKUST-1@CP Composites

A total of 0.3 g Cu-CP was added into 15 mL deionized water and sonicated for 30 min. Meanwhile, the desired amount (according to the Cu contents in Cu-CP at the same molar ratio of synthesis of HKUST-1) of trimesic acid was dissolved into 15 mL ethanol and continually stirred at 50 °C for 30 min. The desired amount of triethylamine was then added into the solution and stirred at 50 °C for 3 h. Finally, the HKUST-1@CP composites were obtained after cooling down, washing, centrifuging, and drying at 120 °C overnight, and named HKUST-1@CP-X-E-Y, correspondingly.

The amounts of the used ligands and the triethylamine were summarized in Appendix A, and the schematic diagram for CP preparation was shown in Appendix A of the Appendix A.

### 2.6. Characterizations

XRD patterns were recorded using the Beijing Purknje General Instrument Corporation XD-6 X-Ray diffractometer with Cu Kα for 4°·min^−1^ in 2 *θ* of 5–50° at 36 kV and 20 mA. According to the relative value of the sum of ten diffractive peak intensities in the XRD patterns of the synthetic CPs, namely, indexed as (020), (200), (111), (13–1), (131), (22–2), (42–2), (350), (530), and (061), the crystallinity of the CP synthesized with the crystallized time of 72 h at 150 °C without any additive PEG and normalized as 100%. The relative crystallinity of the other samples could then be calculated on the basis of Equation [25], as follows (1):(1)$$Crystallization\;degree\;\left(\%\right)=\frac{\sum_{i=1}^{10}I_{i}}{\sum_{j=1}^{10}I_{j}}\times 100\%$$
where $\sum_{i=1}^{10}I_{i}$ is the sum of the intensities of the ten peaks of the XRD pattern of the samples synthesized with different crystallization time and $\sum_{j=1}^{10}I_{j}$ which refers to the sum of the intensities of ten peaks of the XRD pattern of the CP. *I* is the intensity of each diffractive peak, and *i* or *j* is a number from 1 to 10, corresponding to diffractive peaks, indexed as (020), (200), (111), (13–1), (131), (22–2), (42–2), (350), (530), and (061), respectively.

The metal ion contents were determined using a Hitachi ZA3300 Polarized Atomic Absorption Spectrometer (AAS). The samples were dissolved in HNO_3_ solution as follows: 0.02 g sample was dissolved in 0.5 mL 23 mol/L HF solution, and then diluted to 10 mL with HNO_3_ (Volume fraction 2%). Subsequently, 1 mL of the above solution was diluted to 10 mL with HNO_3_ (Volume fraction 2%). The morphologies were obtained using SEM (JEOL JEM-220) images with the microscopes at 15.0 and 200 kV and the energy dispersive X-ray spectroscopy (EDS). The IR-Prestige-21 FT-IR spectrum was used to measure the functional groups of the obtained samples in the wavenumber range of 400–4000 cm^−1^. The TG profiles were received using Perkin–Elmer Pyris Instruments TG–DSC thermal analyzer at a heating rate of 10 °C·min^−1^ and a N_2_ flow rate of 20 mL·min^−1^. The N_2_ adsorption–desorption isotherms at 77 K as well as the CO_2_ and CH_4_ adsorption at 273 K and 298 K were both collected using a Beijing JWGB JW–BK300 gas adsorption instrument. Each sample was outgassed at 120 °C for 6 h before adsorption. The Small Angle X-ray Scattering (SAXS) patterns were performed at the 1W2A station of the Beijing Synchrotron Radiation Facility with the wavelength of the X-ray source of 0.154 nm. The distance from sample to detector was 1600 mm, which was the calibrated reference of the diffraction ring of a standard sample. The scattering vector magnitude was around 0.09 to 3.04 nm^−1^, and the detector readout noise (dark current) of the Mar165 CCD measured with a mask before the sample measurements was approximately 10 counts per second. The sample was sealed with Scotch tape into a sample cell with 1 mm thickness. The scattering images were collected through the single–frame mode with a “multi-read” of 2 times, and the exposure time was 5 min. The two-dimensional SAXS images were transformed to the one-dimensional data by the Fit2D software (http://www.esrf.eu/computing/scientific/FIT2D, accessed on 25 January 2016) and the S program package [26].

### 2.7. Breakthrough Experiments

The breakthrough experiments were tested by a BSD-MAB Multi-Component Adsorption Breakthrough Curve Analyzer containing a cylindrical quartz column, a mass flow controller, pressure-control valves, and a mass spectrograph that was used to evaluate the dynamic separation performance of HKUST-1@CPs. The two adsorbing gases had a ratio of 50/50 vol%, and they were introduced into the controlled environment with the total flow rate of 5 mL/min. The desorption gas flow was detected with a BSD-MAB multi-component mass spectrograph. The breakthrough column with the bore size of 4.0 mm containing the adsorbate sample of about 0.1 g was installed inside the ceramic oven, which was located inside the convection oven. The pressure drops across the column and in the outlet of the column were both monitored. The gas flow sent to the analysis section could be adjusted with the valve. The sample was outgassed at 393 K for 2 h under a nitrogen atmosphere with the flow rate of 20 mL/min, and it was then cooled down before the breakthrough experiment.

## 3. Results and Discussion

### 3.1. Structure Characterizations

Figure 1 shows the XRD patterns of the HKUST–1@CPs, and the synthetic CPs and Cu–CPs are shown in Appendix A. As can be seen in Appendix A, both the CP-0 and the CP-24 presented diffractive peaks that were not obvious when the crystallization time was less than 24 h, while Appendix A shows that CP-72 exhibited the same intensively diffractive peaks as the conventional CP [27], namely, indexed as (020), (200), (111), (13-1), (131), (22-2), (42-2), (350), (530), and (061), which is indicative of the HEU structures [28]. However, its characteristic peaks (Appendix A) not only slightly decreased in intensity but also shifted to a more or less high region in the 2 theta position, as compared with that of the pure CP (Appendix A). In contrast, the weakened intensity (200) of the CP-72-E (as shown in Appendix A) implied decreases in its long-range order, possibly due to the delamination of the HEU lamellas [29]. Meanwhile, the intensity of other diffractive peaks before and after delaminated CP-72 can be well maintained. As a result, CP lamellas would be disordered in a specific crystal plane after delamination, which would lead to the achievement of delamination. In other words, delamination obviously reduced the long-range order of the CP, and, thus, more surface cations were exposed [29], which may facilitate the formation of MOF with more ligands through coordination in the following section.

Meanwhile, the role of Zn^2+^ in the delamination process was similar to the principle of the reference [16], namely, the formation of ZnO_x_(OH)_y_ in the interlayer of CP. Briefly, a consequence of a single Zn^2+^ inserted interlayer and complete exchange with two Na^+^ or K^+^. A significant amount of Zn^2+^ remained in CP-72-E (0.96 wt%), which is a result of Zn^2+^ intercalation as well as nucleation and growth of small Zn(O)_x_(OH)_y_ colloids in between the layer spaces. Beyond that, the (200) diffractive peak located at around 11.17^o^ in the XRD pattern revealed the presence of a HEU structure. After the delamination in the Zn-containing HNO_3_ solutions, the (200) peak position was not shifted and its corresponding *d* value was 0.79140 nm, indicating the Zn-containing HNO_3_ modification had no significant impact on (200) interplanar spacing.

As can be seen in Appendix A, the not obvious change of the (200) peak position after ammonization and the subsequent Cu^2+^ exchange indicated that the ion exchange process had no significant influence on (200) interplanar spacing. The presence of the extra diffractive peaks at 2 theta of 16.10 and 15.40 suggested that the appearance of the atacamite and botallackite in Cu-CP-0-E-10.5 and Cu-CP-24-E-10.1 (Appendix A) were hardly removed via washing and drying [30,31]. Appendix A shows that the Cu content was around 1.64 mmol/g for Cu-CP-0-E-10.5 and 1.58 mmol/g for Cu-CP-24-E-10.1, which is more than the exchangeable cations (equal to total of Na content and K content), meaning that the Cu cations were partially located at the skeleton outside the CP with the low crystallinity.

Regarding the Cu-CPs with different Cu contents (Appendix A), the position of these diffractive peaks was not a significant change, indicating that the HEU structures of the CPs were well maintained after several ion exchanges. Compared with that of the Cu-CPs with low crystallinity (Appendix A), the extra diffractive peaks of atacamite and botallackite were hardly observed in Appendix A. The largest Cu content was around 1.14 mmol/g for Cu-CP-72-E-7.3, which was less than that of the exchangeable cations. In addition, the crystallinity decreased with an increase in Cu contents (shown in Appendix A), showing 93.3% for Cu-CP-72-E-3.4 and 87.8% for Cu-CP-72-E-7.3.

The XRD patterns of the HKUST-1@CPs with different crystallinity of the CP and different Cu contents are shown in Figure 1. As can be seen in Figure 1a,b, the XRD pattern of HKUST-1@CPs exhibited the structural features of HKUST-1 and the CPs, such as (222) for HKUST-1 and (131) for CP, in which the (222) crystal plane of HKUST-1 (as shown in Appendix A) nearly overlapped with the (200) of CP [32]. Beyond that, the disappearances of the diffractive peaks of atacamite and botallackite occurred in HKUST-1@CP-0-E-10.5 and HKUST-1@CP-24-E-10.1 (Figure 1a,b), compared with the corresponding Cu-CPs (Appendix A). These observations suggested the formation of partial HKUST-1 mixture in the CP with low crystallinity. The diffractive peaks of HKUST-1, meanwhile, were relatively weak, and they cannot even be observed in HKUST-1@CP-72-E, which may be due to its low intensity.

As shown in Figure 1, the crystallinity of the HKUST-1@CPs presented increased tendencies with an increase in the crystallization time of the synthesized CPs, showing 33.9% for HKUST-1@CP-0-E-10.5, 44.4% for HKUST-1@CP-24-E-10.1, and 97.9% for HKUST-1@CP-72-E-3.4, respectively. The crystallinity of HKUST-1@CP-72-E-7.3, meanwhile, decreased to 93.6% compared with that of HKUST-1@CP-72-E-3.4.

Herein, the formula was defined as follows:(2)$$I_{r}=\frac{I_{\left(200\right)}}{I_{\left(131\right)}}\;\;\;\;\;\;\;\;\;$$
where *I* represents the intensity of the diffractive peak, and *I_r_* is the specific value of the intensity of (200) and (131), which denotes the relative intensity of (200).

As a result, the *I_r_* value of CP-72 was 40.0%, lower than that (50.8%) of the pure CP synthesized without additive PEG, implying that the PEG effect was somewhat beneficial to reducing the peak (200) intensity of CP. The interplanar spacing (*d*) value of (200) crystal plane of CP-72 was 0.7934 nm, as well, which was larger than that of the pure CP (0.7921 nm), further verifying that the PEG effect during the synthesis process of CP was conducive to enlarge the interplanar spacing of CP.

We noted that the Zn content determined by AAS was nearly zero for Cu-CP-72-E-7.3, suggesting that the generation of Zn-MOFs was well avoided on HKUST-1@CPs. The interlayer Zn^2+^ were fully exchanged after ammonization and Cu^2+^ exchange, owing to cationic sites of Zn^2+^ exposed to the outside layers.

The abovementioned samples were further characterized via the SAXS patterns, and their scattering curves are shown in Figure 2 and Appendix A. As can be seen, the linearity of the *ln*[*I*(*q*)] versus *ln*(*q*) scattering profile revealed the fractal property of the synthesized CPs [33], whereas the linear range was determined in the range of −1.90 < *ln*(*q*) < 0.00, and their corresponding slopes were between −4 and −3.

Accordingly, Figure 2 and Appendix A indicate that all of the samples possessed surface fractal (*D_s_*) features [34], showing an increasing *D_s_* value from 2.14 for CP-72 (Appendix A) to 2.33 for CP-72-E (Appendix A) before and after delamination, but larger than that of pure CP (2.06, as shown in Appendix A). These results implied that the surfaces of the synthetic CP-72-E became rougher with a higher surface activity [35]. Moreover, an increasing *Ds* value appeared from 2.07 to 2.33 with an increase in the crystallization time from 0 to 24 h (shown in Appendix A), suggesting the transformation from a smooth surface to a rough and open structure for the resultant CP [10]. After that, the *Ds* value decreased to 2.14 when the crystallization time was prolonged to 72 h (Appendix A). Similar results were demonstrated by Zhao et al. [36], indicating that the fractal structures were strongly related to the surface property of aluminosilicate species, which supplied active precursors or colloidal nuclei for promoting crystal growth of the CPs. Radlinski et al. [37] demonstrated that the scattering occurred predominantly between the structure of the sheets and the interlayered matters, while the scattering-derived porosity confirmed a minor contribution from micropores.

As can be seen, the *Ds* value increased from 2.07 for CP-0 (Appendix A) to 2.10 for HKUST-1@CP-0-E-10.5 (Figure 2a), and from 2.33 for CP-24 (Appendix A) to 2.40 for HKUST-1@CP-24-E-10.1 (Figure 2b). Meanwhile, the scattering profiles of Cu-CPs with different Cu contents are shown in Appendix A. As can be seen, the *Ds* value increased with the increase in the Cu contents, such as 2.38 for Cu-CP-72-E-3.4 (Appendix A) and 2.44 for Cu-CP-72-E-7.3 (Appendix A). Obviously, these results suggested the successful Cu^2+^-incorporation on CP and further coordination with the ligands. The variation trend of the *Ds* value of HKUST-1@CPs with different Cu contents was similar to that of the Cu-CPs (as shown in Appendix A), 2.53 for HKUST-1@CP-72-E-3.4 (Figure 2c), and 2.60 for HKUST-1@CP-72-E-7.3 (Figure 2d).

The PDDF curves of all related samples are shown in Figure 3 and Appendix A. As shown in Appendix A, the PDDF curves of various CPs lacked the perfect symmetry, particularly that of CP-72 (Appendix A), indicating that their particles probably had flake-like morphologies, which may be related to the thinning of the lamellar thickness [38,39]. However, the PDDF curves of the CPs with a short crystallization time of less than 24 h (as shown in Appendix A had relatively poor symmetry compared with the pure CP (Appendix A) and CP-72 (Appendix A), possibly due to the formation of the flake-like particles.

The PDDF curves of HKUST-1@CPs with a different crystallization degree of CP and different Cu contents are shown in Figure 3. As can be seen, the symmetry of PDDF profiles of the HKUST–1@CPs were almost the same as that of the synthetic CP (as shown in Appendix A), suggesting that the morphologies of CP were not significantly varied during the ammonization Cu^2+^ exchange process and even during the formation of HKUST-1.

The PDDF curves of Cu-CPs with different Cu contents, such as Cu-CP-72-E-3.4, and Cu-CP-72-E-7.3, as shown in Appendix A, show almost the same symmetries of each sample with the same Cu content, namely, HKUST-1@CP-72-E-3.4 and HKUST-1@CP-72-E-7.3, as shown in Figure 3. These results indicate that the ion exchange and the coordinated process had a not obvious effect on the morphologies of the lamination but that they remarkably influenced the layered stacking mode.

The FT-IR spectra of all related samples in the scanned regions between 400 and 4000 cm^−1^ are shown in Appendix A. As can be seen in Appendix A, the bands appeared at 1638, 1205, 1062, 793, 605, and 465 cm^−1^, and were attributed to the features of the synthetic CPs as follows: the band at 1638 cm^−1^ was due to deformation vibrations of H_2_O molecules [40]. The peaks around 1062 cm^−1^ with a shoulder at 1205 cm^−1^ were assigned to the asymmetric internal T-O-T (T = Si or Al) stretching vibrations of the tetrahedral [41]. The band around 465 cm^−1^ was associated with internal T-O bending vibrations of the tetrahedral, while the band located at 605 cm^−1^ was associated with an external tetrahedral double ring [42]. Meanwhile, the peak that appeared at 793 cm^−1^ represented the stretching vibration modes of the O–T–O groups [41], suggesting the presence of the quartz impurity. Appendix A showed that no absorption peaks at 1205, 1062 and 605 cm^−1^ were observed within the crystallization time of 0 and 24 h, indicating the absence of the tetrahedral structures in a short crystallization time. The FT-IR spectra of CP-72-E (Appendix A) was the same as that of CP-72 (Appendix A), meaning that the skeletal structure of the CP was well maintained after delamination and even by acid washing.

As can be seen in Appendix A, the HKUST-1 had obvious adsorption peaks in the ranges of 1700–1500 cm^–1^ and 1500–1300 cm^−1^, which are attributable to the asymmetric and symmetric stretching modes of the carboxylate functional groups, respectively. In particular, the band at 1451 cm^−1^ was related to the stretching and deformation modes of the benzene ring, while the C-H bending mode of the benzene ring appeared at around 729 cm^−1^ [43].

The HKUST-1@CP-72-E-7.3 (Appendix A) had the additional weak signals at around 1451 and 729 cm^−1^, belonging to Cu-O stretching vibration, in which the oxygen atom of the ligand was coordinated with the Cu^2+^ [44]. As shown in Appendix A, the Cu-CP-72-E-7.3 had no adsorption peaks around 1500 cm^−1^, and other signals of the CP were almost the same. However, the band at around 1400 cm^−1^ indicated the unexchanged NH_4_^+^ of the CP [45]. A weak adsorption peak at around 1400 cm^−1^ of Cu-CP-72-E-7.3 (Appendix A) still appeared, as compared with that the CP-72 (Appendix A). These observations implied that a fraction of NH_4_^+^ can hardly be exchanged, although the Cu^2+^ exchange reached the maximum exchange capacity. On the basis of the abovementioned descriptions, the HKUST-1@CP composites were successfully synthesized.

The N_2_ adsorption-desorption isotherms of all the related samples are shown in Appendix A. As can be seen, the isotherms of all the CPs exhibited the characteristic type-I curves with an H3-type hysteresis loop at 0.80 < *P/P_0_* < 0.98, indicating the appearances of the silt–shaped mesopores, probably originating from the lamellar accumulation of the CPs [46,47]. Meanwhile, the specific surface area of CP-72 (Appendix A) was around 33.9 m^2^/g, lower than that (42.0 m^2^/g) of pure CP (Appendix A), but almost same as that (35.9 m^2^/g) of CP-72-E (Appendix A) because the introduction of the additive PEG in the synthesis system was beneficial to expanding the lamellar structures of the synthesized CP, while the effect of the delamination process was unobvious. As can be seen in Appendix A, all of the Cu-CPs presented almost same the isotherm profiles as CP-72-E (shown in Appendix A). The HKUST-1 (shown in Appendix A) showed the adsorption capacity plateau close to zero, indicating the non–porous nature of the sample. The adsorption equilibrium was reached at low pressure, and the hysteresis loop did not occur at high pressure [48], indicating type I isotherm.

As can be seen in Appendix A, the HKUST-1@CP-72-E-7.3 showed the same features as Cu-CP-72-E-7.3 (shown in Appendix A), indicating that the nanopore structures of CP in the composite were well maintained. The BET surface area and the porosity properties of the samples are collected in Appendix A, demonstrating that HKUST-1@CP-0-E-10.5 and HKUST-1@CP-24-E-10.1 (Appendix A) showed a higher specific surface area and pore volume than HKUST-1@CP-72-E-7.3 (Appendix A). Similarly, the specific surface area and pore volume of HKUST-1@CP-24-E-10.1 were lower than HKUST-1@CP-0-E-10.5. Meanwhile, the HKUST-1 presented a higher surface area (516.1 m^2^/g) and total pore volume (0.28 cm^3^/g) [49]. Comparably, the surface area of the HKUST-1@CP-72-E-7.3 was about 55.2 m^2^/g, higher than that (39.5 m^2^/g) of Cu-CP-72-E-7.3 due to the incorporation of HKUST-1 onto CP [50,51]. However, Appendix A shows that the surface area of HKUST-1@CP-72-E-3.4 was lower than that of Cu-CP-72-E-3.4, and the possible reason for this is the formation of the sub-units of HKUST-1 as well as the blockage of the micropores in the CP.

### 3.2. Crystallization Kinetics of the Synthesized CP with the Additive PEG

In order to explore the role of the additive PEG in the synthesis of the CPs in detail, the effects of the crystallization temperature and the time on the crystalline phase are further elucidated via crystallization kinetics. On the basis of the crystallinity of the CP synthesized with and without additive PEG at different temperatures, the crystallization kinetics performances of the related samples synthesized at 140, 150, and 160 °C are shown in Appendix A. As can be seen, there were three distinct regions in the crystallization kinetics curves, namely: (a) an induction period including the nucleation phase, (b) a growth period involving the rapid growth of crystallites, and (c) a stable period comprising the deceleration of the growth process, similar to the reported demonstration [52]. Herein, the induction time (*t*_0_) is defined as the crystallization time that has elapsed in order to achieve the crystallization degree of about 15% [53], as summarized in Appendix A. It should be noted that a crystallinity of 15% was taken into account when determining the induction period [54]. Accordingly, the *t*_0_ value of the synthesized CP with additive PEG gradually decreased with the enhanced temperature from 50 h at 140 °C (Appendix A) to 24 h at 150 °C (Appendix A), and 12 h at 160 °C (Appendix A). Similar phenomena were also observed in the crystallization kinetics of CP synthesized without PEG (Appendix A).

Based on the Arrhenius equation, the activation energy (*E*) during the induction and growth processes was calculated to clarify the crystallization mechanism. The *E* values and frequency factor (*lnA_n_*) for the induction stage (*E_n_*) values are determined from the nucleation rate (1/*t*_0_) and calculated using Equation (3):(3)$$ln\frac{1}{t_{0}}=lnAn-\frac{E_{n}}{RT}$$
where *R* is the ideal gas constant, *T* is the absolute temperature (K), and *A_n_* is the frequency factor for the induced stage [55]. The growth period (*E_g_*) values are also obtained via Arrhenius Equation (4):(4)$$lnkmax=lnAg-\frac{E_{g}}{RT}$$
where *A_g_* is the frequency factor for the growth stage and the growth rate (*k_max_*) is taken as the steepest slope of crystallization curves.

As shown in Appendix A, the *E_n_* values of the synthetic CPs were all larger than their *E_g_* values, suggesting that the nucleation process was a controlled step during the hydrothermal crystallization procedure. Meanwhile, the *E_n_* value of CP-72 synthesized with additive PEG was 106.2 kJ/mol, higher than that (63.0 kJ/mol) of pure CP obtained without PEG, and we can conclude that there is a restraining effect of the PEG on the formation of the crystal nucleus, possibly due to the PEG encapsulation for silicate and aluminate species hindering their polycondensations. Meanwhile, the *E_g_* value was 31.3 kJ/mol for CP-72, less than that for pure CP (59.8 kJ/mol), indicating the promoting effect of the PEG on the growth procedure of the CP.

### 3.3. Morphologies and Thermal Analysis

The morphologies of the related samples are shown in Figure 4. As can be seen in Figure 4b, the dispersive CP precursor solution exhibited the colloidal particles with irregular shapes that had a size of around 70 nm can be assigned to the nanoaluminosilicate species [56]. Subsequently, the particle sizes of CP-24 were increased to 3 μm, besides the appearances of the almost flake-like morphology (Figure 4c). When the crystallization time was further prolonged to 72 h, the random and lamellar accumulations of CP-72 were present with a flower-like morphology in sizes of around 5 μm (Figure 4d), which is a noticeable difference from that of pure CP (Figure 4a).

CP-72 (Figure 4d), meanwhile, had an irregularly lamellar structure in length of 1.5 μm and width of 1.1 μm, which was smaller than that (length of 3.0 μm and width of 2.0 μm) of pure CP (Figure 4a). These results suggested that the PEG effect was useful to shrinking the lamellar size of the CP. Additionally, the lamellar size of CP-72-E (Figure 4e) was 1.2 μm in length and 0.8 μm in width, smaller than that of CP-72 (Figure 4d). The possible reason for this is that the acidity of the Zn-containing HNO_3_ solution caused the dissolutions of the lamellas of the synthetic CPs. Additionally, Figure 4j shows the agglomerated particles of HKUST-1 of a size around 0.5 μm, similar to the result reported by Majano et al. [57]. As can be seen in Figure 4f,g, HKUST-1@CPs presented an obvious HKUST-1 and the amorphous particles, which again confirms demonstrations of XRD patterns (as shown in Figure 1a,b).

The lamellas of CP-72-E (Figure 4e), Cu-CP-72-E-7.3 (Figure 4i), and HKUST-1@CP-72-E-7.3 (Figure 4h) were almost the same size, around 1.0–2.0 μm in length and 0.5–1.5 μm in width. By contrast, unobvious HKUST-1 particles (Figure 4j) appeared in both HKUST-1@CP-24-E-10.1 (Figure 4g) and HKUST-1@CP-72-E-7.3 (Figure 4j). Therefore, we speculated that HKUST-1 may grow mainly along the highly dispersed lamellar surfaces of the CP, which depend on the Cu^2+^ location distributed in CP.

As described in the Experimental section, after each ion exchange process between the CPs and Cu^2+^ was finished, an enormous amount of deionized water was used to wash the obtained solids so as to remove the excess Cu^2+^ cations adsorbed on the CP surfaces. However, the adsorbed Cu^2+^ ions on the low crystallinity CP were still not completely removed, leading to the generations of HKUST-1 deriving from coordination between the ligands and part of Cu^2+^ ions in HKUST-1@CPs (as shown in peak (222) of Appendix A), besides the appearances of atacamite and botallackite in the Cu^2+^-exchanged CPs (as shown in peak (011) of Appendix A and peak (100) of Appendix A) [30,31].

However, HKUST-1 in the HKUST-1@CPs with a high crystallization degree was difficult to observe in the XRD patterns and SEM measurements, and one of the main reasons was the fact that the exchanged Cu^2+^ ions, as the central ion, were located in the equilibrium cation position of the CP, and, therefore, the particle size of the synthesized HKUST-1 was very small. A similar report was made by Wu et al. [58], who synthesized the MOFs@zeolites (NaY and HZSM-5) via coordination between the exchanged Zn^2+^ and the ligands (2-methylimidazole and its derivatives), though the sub-units of the MOFs distributed in Zn-exchanged Zeolites were difficult to observe via XRD patterns, SEM, and TEM measurements.

As described in the Introduction, the delamination process usually follows the synthesis process, which improves the delamination performance. In the present work, the layered structure expansions and the lamellar disorder arrangements of the synthesized CPs were firstly prepared via adding PEG in the hydrothermal synthesis system, and then the delamination process was performed via Zn-containing acid washing under mild conditions.

As can be seen in Appendix A, the weight loss profiles could be divided into three stages: the first one occurred when the temperature was up to 300–500 K, which was attributed to desorption of the physisorbed water, and the second happened during the temperature range of from 500–800 K, which was assigned to the chemisorbed water and possible decompositions of the ligands of HKUST-1 [59]. The weight loss of pure CP (Appendix A), CP-72 (Appendix A), CP-72-E (Appendix A), and Cu-CP-72-E-7.3 (Appendix A) at 300–500 K was 9.1, 8.1, 8.8, and 7.7%, respectively. Those at 500–800 K were 0.6, 0.8, 2.0, and 2.3%, respectively. The last one of nearly 1% at 800–1000 K was due to dihydroxylation and decomposition of the residual ligands of HKUST-1 [60]. These observations suggested that the impacts of the Cu^2+^ exchange and post-treatment (delamination and acid washing) on the thermal stability of the CP were not obvious [61].

The TG curve of HKUST-1, meanwhile, as shown in Appendix A, showed that the first weight-loss stage appeared between 300 and 400 K was 21.8%, due to desorption of the physisorbed water. The second one between 400 and 550 K belonged to the dehydration of hydrated copper cations and the physical desorption of the solvent [60]. The weight loss in the temperature range of around 550–700 K was attributed to decompositions of the ligands of HKUST-1 [59]. In particular, it was found that the weight loss of CP-72, HKUST-1, and HKUST-1@CP-72-E-7.3 at 500–800 K was increased from 0.6% to 2.3%, while that of the first stage (300–500 K) was nearly unexchanged, compared with that of CP-72. Similarly, the weight loss of HKUST-1@CP-72-E-7.3 (Appendix A) at 300–500 K was approximately 8%, almost the same as that of pure CP (Appendix A), and its weight loss of around 2.3% at 500–600 K was ascribed to the dehydration of hydrated copper cations and the physical desorption of the solvent. The weight loss of around 5.5% at 600–1000 K belonged to the decompositions of the ligands of HKUST-1. It is actually difficult for us to verify that the thermal stability of HKUST-1 located on the CP was improved.

As in the abovementioned demonstrations, the crystal size of the synthesized HKUST-1 was very small, resulting in its amount being difficult to quantify. In this regard, the amount of the synthesized HKUST-1 was estimated on the basis of the TG data of Cu-CP-72-E-7.3 (Appendix A), HKUST-1 (Appendix A), and HKUST-1@CP-72-E-7.3 (Appendix A), and the HKUST-1 amount in HKUST-1@CP-72-E-7.3 was estimated to be about 8.3%, much lower than the theoretical value (about 23%, assuming that all of the Cu^2+^ ions distributed on CP were coordinated with ligands). The probable reason for this is that the steric hindrance effect strongly limited the coordination of most of Cu^2+^ ions on CP with the ligands, even though the CP lamellas were disordered after delamination.

The effective mechanism of PEG essentially restrained the formation of the crystal nucleus but promoted the growth procedure of the CPs, leading to the reduction of their peak (200) intensity and the decrease in the lamellar size of the synthesized CP.

### 3.4. CO_2_ and CH_4_ Adsorption Performances

Figure 5, Figure 6, and Appendix A all show the CO_2_ and CH_4_ equilibrium adsorption capacities and isotherms of the related samples. As can be seen in Figure 5A,C, the CO_2_ and CH_4_ adsorption capacity of the HKUST-1@CPs increased gradually with the increase in the crystallinity of the CP, namely, HKUST-1@CP-0-E-10.5 < HKUST-1@CP-24-E-10.1 < HKUST-1@CP-72-E-7.3, showing 0.68, 0.78, and 1.14 mmol/g for CO_2_ at 273 K (Figure 5(Aa–c)) and 0.06, 0.11, and 0.23 mmol/g for CH_4_ at 273 K (Figure 5(Ca–c)), respectively, which is the same phenomena at 298 K as the one shown in Figure 5B,D.

Appendix A presented the CO_2_ and CH_4_ adsorption capacity of the CPs synthesized at 273 K, showing the enhanced tendencies with the increased crystallinity, from 0.80 and 0.02 mmol/g for CP-0 to 1.39 and 0.31 mmol/g for CP-72, respectively. The similar phenomena at 298 K were also shown in Appendix A. Meanwhile, as shown in Appendix A, we noticed that the CO_2_ and CH_4_ adsorption capacity at 1 bar at both 273 K and 298 K was decreased in the following order: pure CP > CP-72 > CP-72-E, suggesting that the effects of the disordered lamellar morphologies of the synthetic CPs on the CO_2_ and CH_4_ adsorption performances were not obvious.

However, as shown in Figure 6A,C, the CO_2_ and CH_4_ adsorption capacity of HKUST-1@CP-72-E-3.4 presented 1.39 and 0.38 mmol/g at 273 K, more than that (1.14 and 0.23 mmol/g) of HKUST-1@CP-72-E-7.3. The similar results at 298 K were also shown in Figure 6B,D. Meanwhile, Appendix A showed that the CO_2_ and CH_4_ adsorption capacity of Cu-CP-72-E-3.4 at 273 K were 1.25 and 0.28 mmol/g, lower than that (0.84 and 0.21 mmol/g) of Cu-CP-72-E-7.3, respectively. The similar observations at 298 K were also shown in Appendix A.

As is well known, the dynamic diameter is 0.38 nm for CO_2_ and 0.33 nm for CH_4_, besides their almost same polarizability. However, the quadrupole of CO_2_ is 4.30 × 10^−26^ cm^2^, while the quadrupole of CH_4_ is almost zero. Obviously, the interaction force of CO_2_ with the synthesized CPs is stronger than that of CH_4_, which is conducive to the enhancements of its adsorption capacity and the improvement of its adsorption selectivity [62]. Meanwhile, Xin et al. proposed the possible CO_2_ adsorption mechanism on HKUST-1 [63], suggesting the appearance of two adsorption sites: one is the small pore cages composed of a ring of six metal dimers and six trimesate groups, and the other is the open Cu ions. Spanopoulos et al. further elucidated that the adsorption performances of CO_2_ mainly occurred at the two sites stated above, while that of CH_4_ only occurred at the open Cu structures [64].

Koyama et al. proposed four kinds of the exchangeable cation positions, namely, M(l), M(2), M(3), and M(4), which are distributed in different channels [65]. In detail, M(l) is located in channel A and is coordinated by oxygen atoms of two frameworks and five water molecules. M(2) appeared in channel B and is adjacent to the oxygen atoms of three frameworks and five water molecules. M(3) centered on channel C is nearly the center of its eight-member ring. Similar to M(l), M(4) appeared in channel A at a center of inversion, and can be coordinated with six water molecules occupying the vertices of an octahedron. The M(4) position may accommodate these excess atoms [65,66,67]. As reported by Garcia-Basabe et al. [68], most of the Cu^2+^ on CP were located in the center of channel A and channel B of two extra framework sites, indicating that the Cu^2+^ presence was mainly distributed at M(1) and M(4). They further demonstrated that an additional new site was found in channel A at distances of 0.145 and 0.165 nm from the M(1) and M(4) sites, originating from the Cu^2+^ coordination with water molecules.

In the present work, our results seem to be related to the Cu^2+^ positions distributed on the synthetic CPs and the adsorption sites, depending on the crystallization of the synthetic CPs and the possibilities of the coordinated HKUST-1.

Additionally, the CO_2_ and CH_4_ adsorption capacities of Cu-CP-72-E-7.3 at 273 K and 298 K (shown in Appendix A) were even lower than that of CP-72-E (shown in Appendix A). These phenomena were mainly because of the hydrochloric acid generation deriving from the Cu^2+^ hydrolysis leading to the destruction of the adsorption sites, which is consistent with the decrease in CP crystallinity that occurs with the increase in Cu content (as shown in Appendix A).

On the basis of the abovementioned results, the CO_2_ and CH_4_ isosteric adsorption heat of all related samples was calculated via the Clausius–Clapeyron equation, seen here in Equation (5):(5)$$ln\frac{P_{1}}{P_{2}}=\frac{\Delta H_{vap}}{R}\left(\frac{1}{T_{1}}-\frac{1}{T_{2}}\right)$$
where *P*_1_ and *P*_2_ represent the pressure under the temperatures of *T*_1_ and *T*_2_, Δ*H_vap_* represents the isosteric adsorption heat of CO_2_, and R represents the gas constant (8.314 J·mol^−1^·K^−1^) [69]. The Freundlich–Langmuir (F-L) equation, which is Equation (6) below, was used to fit the relationship between adsorbed amounts and relative pressure under the same adsorbed amounts:(6)$$Q=\frac{q_{s}KC^{n}}{1+KC^{n}}$$
where *Q* is the adsorbed amount under different relative pressure (mmol/g), *q_s_* is the molar adsorption capacity of the system when the *Q* is nearly equal with an increase in relative pressure (mmol/g), *K* and *n* are F-L constants, and *C* is the relative pressure.

The isosteric heat of CO_2_ and CH_4_ adsorption for various samples is shown in Figure 7. As can be seen there, the isosteric heats of CO_2_ and CH_4_ adsorption were less than zero, indicating that the CO_2_ and CH_4_ adsorption belonged to an exothermic process [70]. In detail, the CO_2_ adsorption heats of HKUST-1@CP-0-E-10.5 (Figure 7(Aa)) and HKUST-1@CP-24-E-10.1 (Figure 7(Ab)) with low crystallinity of the CP were much higher than that of HKUST-1@CP-72-E-7.3 with the high crystallinity (Figure 7(Ad)), while the CH_4_ adsorption heats of HKUST-1@CP-0-E-10.5 (Figure 7(Ba)) and HKUST-1@CP-24-E-10.1 (Figure 7(Bb)) with low crystallinity of the CP were lower than that of HKUST-1@CP-72-E-7.3 with the high crystallinity (Figure 7(Bd)).

The absolute value of the CO_2_ isosteric heat of HKUST-1@CP-72-E-3.4 and HKUST-1@CP-72-E-7.3 (shown in Figure 7(Ac,d,Bc,d)) with different Cu contents was higher than that of CH_4_, indicating a stronger CO_2_ adsorption. Obviously, these results suggest that the CO_2_ adsorption is preferential compared with that of CH_4_. Meanwhile, the absolute values of the isosteric heat of all samples were less than 40 kJ/mol, indicating that the behaviors of the CO_2_ or CH_4_ adsorption belonged to the physical adsorption [71].

To further verify the separation performance of CO_2_/CH_4_ of the samples, the selectivity at 273 K and 298 K was shown in Figure 8. As can be seen in Figure 8A, HKUST-1@CP-0-E-10.5 and HKUST-1@CP-24-E-10.1 with low crystallinity of CP (Figure 8(Aa,b)) showed higher selectivity owing to their low CH_4_ adsorption capacity, but their low CO_2_ adsorption capacity may limit their applications in CO_2_ and CH_4_ separation. The similar results of their CO_2_/CH_4_ selectivity at 298 K were also shown in Figure 8(Ba,b).

Meanwhile, the effect of the coordination of HKUST-1 with Cu-CPs on the separation performance of CO_2_/CH_4_ and the corresponding selectivity at 273 K and 298 K was evaluated. As shown in Figure 8(Ac–e,Bc–e), the selectivity of HKUST-1@CP-72-E-3.4 was lower than that of HKUST-1@CP-72-E-7.3 but higher than that of the prepared HKUST-1 at both 273 K and 298 K. Obviously, one of the main reasons for this is due to the generation of HKUST-1 in the HKUST-1@CPs, with an increase in the Cu content. The CO_2_/CH_4_ selectivity of various samples at different temperatures under the pressure of 1 bar was collected in Table 1. Comparably, the reported selectivity of HKUST-1 depended on the prepared methods [24,59,72].

### 3.5. Breakthrough Performance

Taking HKUST-1@CP-72-E-7.3 as an example, Figure 9 showed the breakthrough profiles of the binary mixtures (CO_2_ and CH_4_) at 50% CO_2_ concentration (by volume) through the fixed bed packed with the sample pellets at 298 K. As can be seen, the relationships between *C*/*C*_0_ versus time, where *C* and *C*_0_ were the volumetric concentration of CO_2_ or CH_4_ in the outlet and inlet stream, showed that the purified CH_4_ was detected in the outlet gas flow but that CO_2_ was still trapped in the column when the mixture of CO_2_ and CH_4_ flowed into all of the column, thereby showing the highly adsorbed CO_2_ performance of the prepared HKUST-1@CP as well as the adsorptive separation properties between the CH_4_ and CO_2_ [73,74].

## 4. Conclusions

The HKUST-1@CPs with randomly lamellar morphologies were successfully synthesized via the PEG-additive hydrothermal route and the Cu^2+^ coordinated routes with trimesic acid. Various characterizations demonstrated that the additive PEG had a strong hindering effect on the formation of the crystal nucleus of CPs but a promoting effect on their growth, although their nucleation process was a controlled step in the crystallization duration. HKUST-1@CP with various Cu^2+^ contents or different crystallizations presented the enhancements of the adsorptive separation for CO_2_ and CH_4_ as compared with Cu^2+^ exchanged CPs and synthesized HKUST-1. In particular, the higher CO_2_/CH_4_ selectivity and better breakthrough performances demonstrated that the obtained HKUST-1@CPs may be a good candidate for further study on potential applications in gas separation.

## Figures and Tables

**Figure 1 nanomaterials-13-01860-f001:**
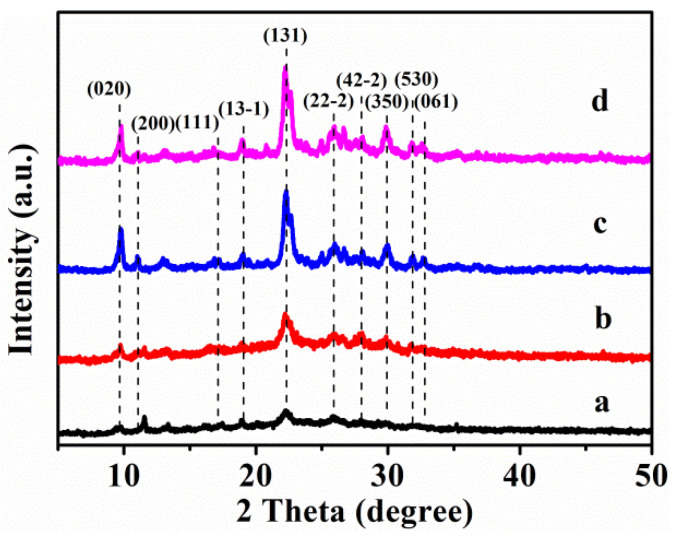
XRD patterns of (**a**) HKUST-1@CP-0-E-10.5, (**b**) HKUST-1@CP-24-E-10.1, (**c**) HKUST-1@CP-72-E-3.4, and (**d**) HKUST-1@CP-72-E-7.3.

**Figure 2 nanomaterials-13-01860-f002:**
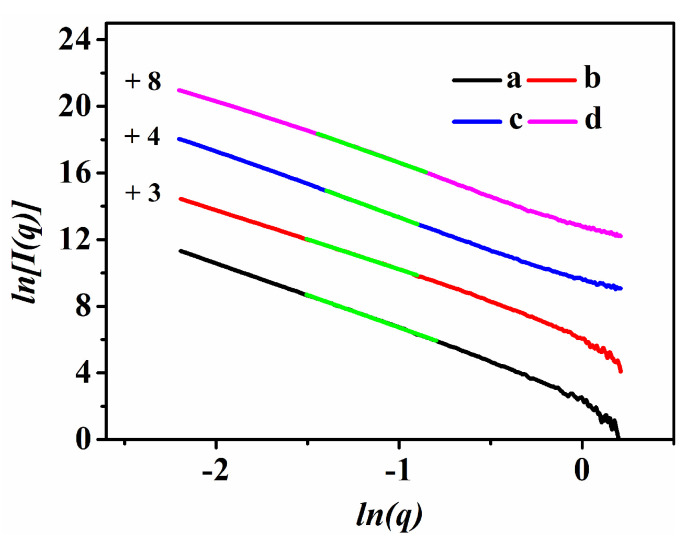
SAXS patterns of (**a**) HKUST-1@CP-0-E-10.5, (**b**) HKUST-1@CP-24-E-10.1, (**c**) HKUST-1@CP-72-E-3.4, and (**d**) HKUST-1@CP-72-E-7.3.

**Figure 3 nanomaterials-13-01860-f003:**
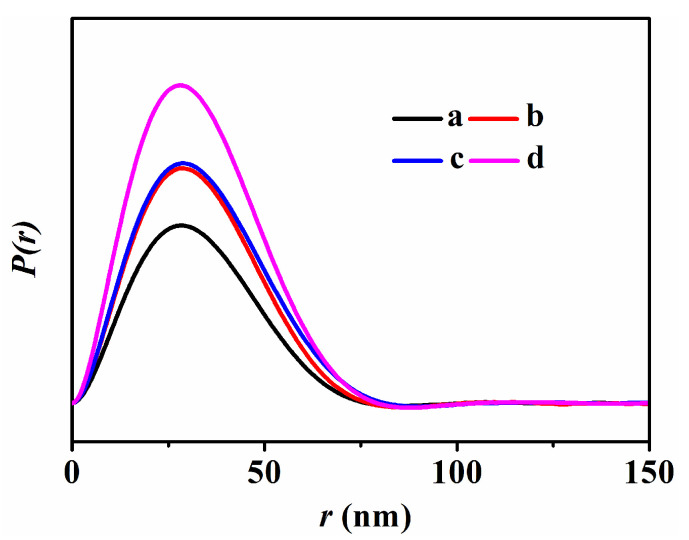
The pair distance distribution function (P(r)~r) profiles of (**a**) HKUST-1@CP-0-E-10.5, (**b**) HKUST-1@CP-24-E-10.1, (**c**) HKUST-1@CP-72-E-3.4, and (**d**) HKUST-1@CP-72-E-7.3.

**Figure 4 nanomaterials-13-01860-f004:**
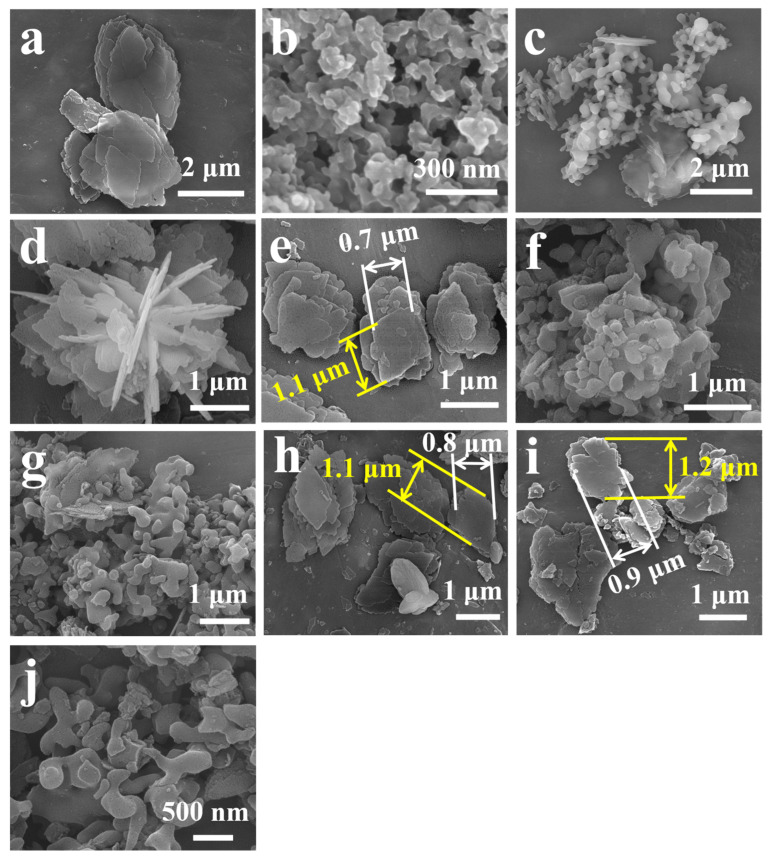
SEM images of (**a**) pure CP, (**b**) CP-0, (**c**) CP-24, (**d**) CP-72, (**e**) CP-72-E, (**f**) HKUST-1@CP-0-E-10.50, (**g**) HKUST-1@CP-24-E-10.08, (**h**) HKUST-1@CP-72-E-7.33, (**i**) Cu-CP-72-E-7.33, and (**j**) HKUST-1.

**Figure 5 nanomaterials-13-01860-f005:**
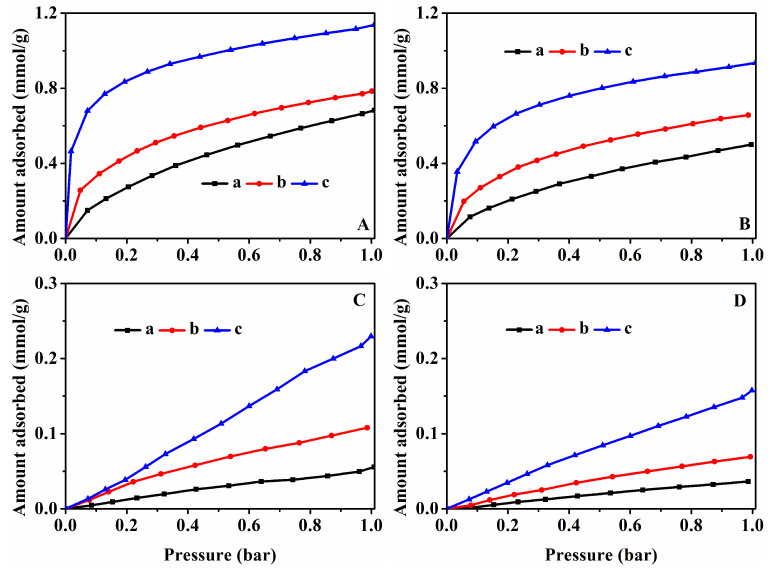
Equilibrium adsorbed isotherms of the samples using CO_2_ as adsorbate at 273 K (**A**) and 298 K (**B**), respectively; CH_4_ as adsorbate at 273 K (**C**) and 298 K (**D**), respectively: (**a**) HKUST-1@CP-0-E-10.5, (**b**) HKUST-1@CP-24-E-10.1, and (**c**) HKUST-1@CP-72-E-7.3.

**Figure 6 nanomaterials-13-01860-f006:**
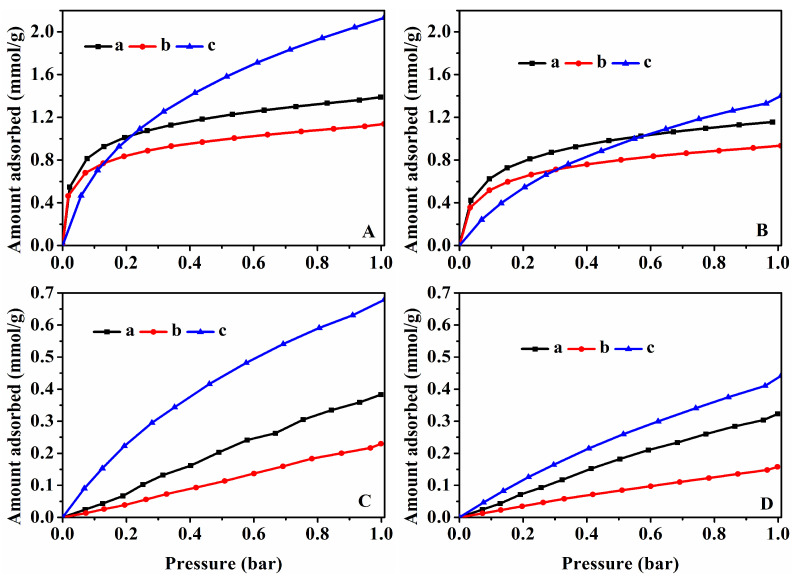
Equilibrium adsorbed isotherms of the samples using CO_2_ as adsorbate at 273 K (**A**) and 298 K (**B**), respectively; CH_4_ as adsorbate at 273 K (**C**) and 298 K (**D**), respectively: (**a**) HKUST-1@CP-72-E-3.4, (**b**) HKUST-1@CP-72-E-7.3, and (**c**) HKUST-1.

**Figure 7 nanomaterials-13-01860-f007:**
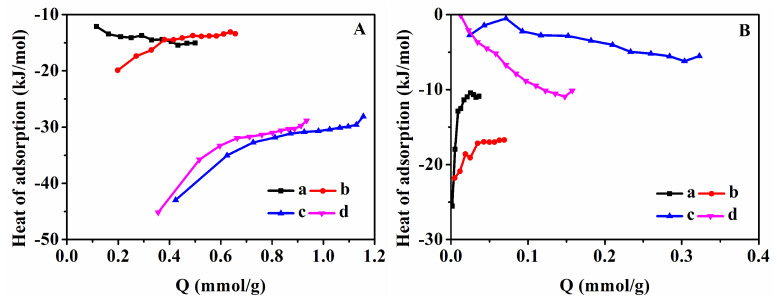
CO_2_ adsorption heat (**A**) and CH_4_ adsorption heat (**B**) of (**a**) HKUST-1@CP-0-E-10.5, (**b**) HKUST-1@CP-24-E-10.1, (**c**) HKUST-1@CP-72-E-3.4, and (**d**) HKUST-1@CP-72-E-7.3.

**Figure 8 nanomaterials-13-01860-f008:**
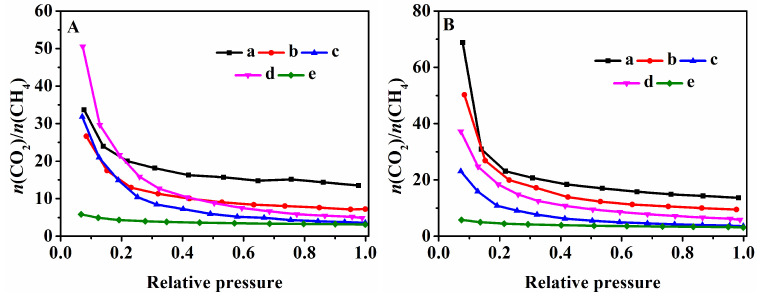
CO_2_/CH_4_ selectivity at 273 K (**A**) and 298 K (**B**) of (**a**) HKUST-1@CP-0-E-10.5, (**b**) HKUST-1@CP-24-E-10.1, (**c**) HKUST-1@CP-72-E-3.4, (**d**) HKUST-1@CP-72-E-7.3, and (**e**) HKUST-1.

**Figure 9 nanomaterials-13-01860-f009:**
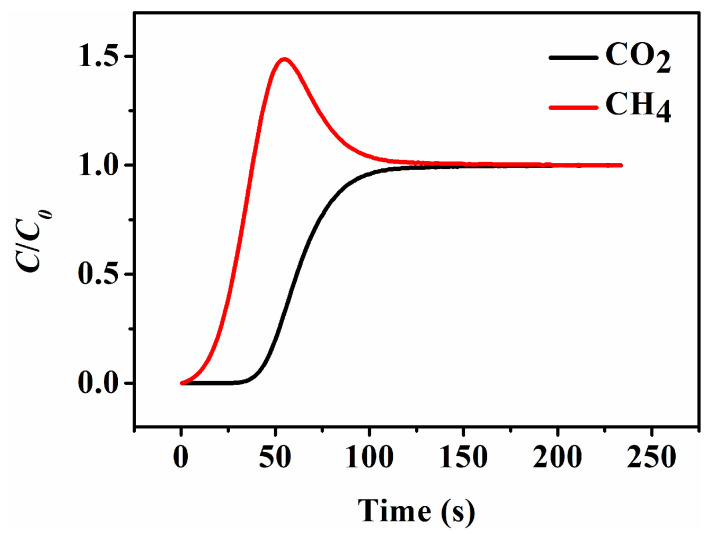
Breakthrough curves of a flowing mixed gas of CO_2_/CH_4_ of 50: 50 vol% through a packed bed of HKUST-1@CP-72-E-7.3 at 298 K.

**Table 1 nanomaterials-13-01860-t001:** Summaries of CO_2_/CH_4_ selectivity of various samples at different temperatures under pressure of 1 bar.

Sample	Temperature	CO_2_/CH_4_ Selectivity
HKUST−1@CP−0−E−10.5	273 K 298 K	13.51 13.65
HKUST−1@CP−24−E−10.1	273 K 298 K	7.23 9.48
HKUST−1@CP−72−E−3.4	273 K 298 K	3.58 3.57
HKUST−1@CP−72−E−7.3	273 K 298 K	4.89 5.87
HKUST−1	273 K 298 K	3.13 3.12
HKUST−1 ^1^ cited in reference [72]	273 K	7.19
HKUST−1 ^2^ cited in references [24,59]	273 K	4.00

^1^ HKUST–1 synthesized by hydrothermal method. ^2^ HKUST–1 synthesized at room temperature.

## Data Availability

The data presented during the study is available on request from the corresponding author.

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
