# Peer review of "Constructing Randomly Lamellar HKUST–1@Clinoptilolite through Polyethylene Glycol—Assisted Hydrothermal Method and Coordinated Complexation for Enhanced Adsorptive Separation for CO2 and CH4"

_nanomaterials, 2023, doi:10.3390/nano13121860_

Round 1

Reviewer 1 Report

The authors present a novel and efficient method for preparing composites of clinoptilolite (CP) and HKUST-1 using a polyethylene glycol (PEG)-assisted hydrothermal route. The composites' structural and physicochemical properties were thoroughly characterized using a variety of techniques, such as XRD, FT-IR, TG, N2 sorption isotherms, and SEM. The study highlights the impact of PEG additives on the formation and growth of CP crystal nuclei.

The paper's primary strength lies in its clear demonstration of the improved adsorptive separation of CO2 and CH4, higher selectivity, and better breakthrough performance offered by the HKUST-1@CP composites compared to individual components. These results showcase the potential of HKUST-1@CP composites as promising candidates for gas separation applications, particularly in CO2/CH4 separation.

The manuscript is well-organized and clearly written, making it accessible to readers in the field of materials science and gas separation. The methodology employed is sound, and the authors have successfully addressed the topic's complexities. Furthermore, the work significantly contributes to the development of advanced materials for gas separation and adsorption applications.

In conclusion, I believe that the publication of this paper will be a valuable addition to the journal and provide meaningful insights to researchers working on gas separation and adsorption materials. I strongly recommend the publication of this manuscript.

Author Response

Response to Reviewer 1#

Reviewer 2 Report

Zhang et al. have reported on the synthesis of a HKUST–1@clinoptilolite composite material for CO2/CH4 separation. However, the manuscript may not benefit the readers in this field until certain issues concerning the preparation of materials and adsorption testing are addressed. Therefore, I would recommend to reconsider this manuscript after a major revision.

1. First of all, there is a lack of evidence for the successful formation of HKUST-1 within the clinoptilolite matrix:

1) The XRD results do not show typical peaks for the formation of HKUST-1, and it is possible that metal-organic clusters were formed between Cu and the ligands, rather than integrated MOF crystals.

2) SEM images do not clearly indicate the existence of MOF morphologies that can be unambiguously distinguished.

3) In table S2, why is BET surface area of [email protected] lower than that of Cu-CP-72-E-3.4 if MOF, the high-surface area component, was formed successfully?

2. the authors quantified the Cu ratio in each composite material but missed the relative quantity of MOF. If the authors assume that all Cu ions are ligated with trimesic acid, evidence needs to be provided.

3. It is necessary for the authors to reconsider the scientific meaning and superiority of assembling HKUST-1 MOF and clinoptilolite as a composite material since the CO2/CH4 adsorption test indicates that the incorporation of clinoptilolite into HKUST-1 impedes the uptake of CO2 or CH4. The authors should emphasize the enhanced performance and properties that can be obtained from mixing up the MOF and zeolite as a composite material and

4. The authors need to discuss the roles of MOF and zeolite in the separation of CO2 and CH4 and how each part interacts with gas molecules.

5. The manuscript highlights the important role of PEG as surfactants for the preparation of clinoptilolite in the title and conclusion. However, no control experiments were conducted to verify this role, and no comprehensive discussion was included in the main text.

Below are some minor concerns:

6. Paragraph 1, ‘conductive’ is confusing, please clarify the exact meaning that the authors want to convey;

7. The authors compiled many examples about delamination methodologies (ultrasonic, calcination etc) used in zeolite into two big paragraphs from line 61 to 105 which is redundant and need to be streamlined. The authors did not convey how these strategies are inspiring and relevant to the delamination method used in this work as well.

8. Are Cu2+ ions adsorbed or exchanged in CP? 

9. Line 114, is the thermal stability of HKUST-1 in this composite material increased? If not, the statement here is confusing.

10. Please separate MOF synthesis and composite synthesis in the experimental part.

11. Please specify ‘desired amount’ in either the experimental part or supporting information. A table listing the Cu quantity and amount of ligand used will help.

12. Line 200-201, what is the temperature for the 30-min and 3-h stirring respectively, since cool down was mentioned later in line 202?

Author Response

Response to Reviewer 2#

Reviewer 3 Report

Introduction should be reduced focusing on the aim of the research.

Many typo and grammatical errors should be corrected. (line 108; 120, 122....) 177; After being washed with... 313..possibly due..

2.2

Reviewer thinks moisture effects are important for the synthesis of intermideates and CP...

Solid chemicals were used; was aluminium hydroxide which is not dissolved in water used for a supporter? A schematic diagram for the preparation is required. 

line 175-176; Instead of 'the above mentioned solution', a new ID synthesized from the process should be presented. 

178; several hours should be rewritten with an apparent hour.

173-190; Was synthesis and delamination carried out separately in this exp? any reasons?

256-271; What is Figure S1 in here?

Description of Figure 1 is not sufficient to comprehend, and the reason why more exposure of surface cations should be discussed.

 272-281; A core parameter for crystal growth is Zn? Theoretical background should be explained even with references. 

327-348; does surface fractal dominate the rx rather than micropore clusters?It could be possible by porosity analysis: 10.1016/j.coal.2004.03.002

p. 11; Details of SEM analysis should be provided including magnification. XRD results could be compared with the SEM. 

p.12; TGA also should be described in details. Weight loss due to moisture also should be discussed. 

Author Response

Response to Reviewer 3#

Round 2

Reviewer 2 Report

My concerns have been addressed.

Author Response

Response to Reviwer 2#

Reviewer 3 Report

Abstract

large pore volume and specific surface area

-> better to present numeric numbers, and pore size is more critical than area or volume for CO2 molecule adsorption, isn't it?

Introduction

The originality of the study is ambiguous. Pls make a clear description.

214;

An adsorption test set-up should be presented including experimental parameters such as sorbent amount, concentration and flow rate.

Results

Fig 1 XRD patterns...do not obvious tendency of each crystalline. It needs more precise discussion. 

287; is eq (2) representing the concentration or composition of each compounds?

 Fig 9; The test  lasted only 4 minutes. Please show the results of longer adsorption experiments, if any, so that the results can be trusted.

There are too many references for a single paper, pls delete some which are not directly related.

Since the discussion for the various analysis results is too lengthy, it should be summarized focusing on the intended theory.

Author Response

Response to Reviewer 3#
